# Characterizing Complex Spatiotemporal Patterns from Entropy Measures

**DOI:** 10.3390/e26060508

**Published:** 2024-06-12

**Authors:** Luan Orion Barauna, Rubens Andreas Sautter, Reinaldo Roberto Rosa, Erico Luiz Rempel, Alejandro C. Frery

**Affiliations:** 1Applied Computing Graduate Program (CAP), National Institute for Space Research, Av. dos Astronautas, 1.758, Jardim da Granja, São José dos Campos 12227-010, SP, Brazil; rubens.sautter@gmail.com (R.A.S.); reinaldo.rosa@inpe.br (R.R.R.); 2Laboratory for Computing and Applied Math, National Institute for Space Research, Av. dos Astronautas, 1.758, Jardim da Granja, São José dos Campos 12227-010, SP, Brazil; 3Mathematics Department, Aeronautics Institute of Technology, Praça Marechal Eduardo Gomes, 50, Vila das Acácias, São José dos Campos 12228-900, SP, Brazil; erico.rempel@ita.br; 4School of Mathematics and Statistics, Victoria University of Wellington, P.O. Box 600, Wellington 6140, New Zealand; alejandro.frery@vuw.ac.nz

**Keywords:** nonlinear dynamics, spatiotemporal patterns, turbulence, Shannon entropy, Tsallis entropy, gradient pattern analysis

## Abstract

In addition to their importance in statistical thermodynamics, probabilistic entropy measurements are crucial for understanding and analyzing complex systems, with diverse applications in time series and one-dimensional profiles. However, extending these methods to two- and three-dimensional data still requires further development. In this study, we present a new method for classifying spatiotemporal processes based on entropy measurements. To test and validate the method, we selected five classes of similar processes related to the evolution of random patterns: (i) white noise; (ii) red noise; (iii) weak turbulence from reaction to diffusion; (iv) hydrodynamic fully developed turbulence; and (v) plasma turbulence from MHD. Considering seven possible ways to measure entropy from a matrix, we present the method as a parameter space composed of the two best separating measures of the five selected classes. The results highlight better combined performance of Shannon permutation entropy ($S_{H}^{p}$) and a new approach based on Tsallis Spectral Permutation Entropy ($S_{q}^{s}$). Notably, our observations reveal the segregation of reaction terms in this $S_{H}^{p}\times S_{q}^{s}$ space, a result that identifies specific sectors for each class of dynamic process, and it can be used to train machine learning models for the automatic classification of complex spatiotemporal patterns.

## 1. Introduction

The intricate relationship between probability and entropy is a cornerstone in information theory and statistical thermodynamics, providing a robust framework for analyzing a multitude of phenomena ranging from data transmission processes to the behavior of many physical systems. Entropy, derived from the probability distribution of the states of a process or system, can be interpreted as a quantitative measure of randomness or disorder, offering deep insights into the underlying dynamics of several complex systems (see, for instance, Refs.  [1,2,3,4,5,6]).

From a thermodynamic perspective, the entropy concept is intimately tied to the statistical mechanics of microstates. Entropy, *S*, is defined by Boltzmann’s entropy equation, $S=k_{B}\ln\Omega$, where $k_{B}$ is the Boltzmann constant and Ω represents the number of microstates. This relationship can be interpreted as the degree of disorder or randomness in a system’s microscopic configurations, drawing a direct connection between the macroscopic observable properties and the statistical behavior of microstates. Complementarily, in the realm of information theory, entropy is fundamentally concerned with quantifying the expected level of “information”, “surprise”, or “uncertainty” in the potential outcomes of a system [7]. This quantification is intricately linked to the probability distribution of these outcomes. It essentially measures the average unpredictability or the requisite amount of information needed to describe a random event, thereby providing a metric for the efficiency of data transmission and encoding strategies. Therefore, the duality of the entropy interpretation works as a bridge between the abstract realm of information and the tangible world of the statistics of physical systems. It encapsulates the essence of entropy as a fundamental measure, providing a unifying lens through which the behavior of complex systems, whether in the context of information processing or thermodynamics, can be coherently understood and analyzed. This interdisciplinary approach not only deepens our understanding of individual phenomenon but also reveals the underlying universality of the concepts of randomness and information across diverse scientific domains.

In the scenario described above, it is necessary to identify entropy measures that are effective in characterizing the spatiotemporal patterns of complex processes typically observed or simulated in 3D+1: following the notation of the amplitude equation theory, where *D* corresponds to the spatial dimension in which the amplitude of a variable fluctuates over time. This need is justified by the great advances in the generation of big data in computational physics, with emphasis on the direct numerical simulation (DNS) of turbulence [8,9], ionized fluids [10,11,12,13,14], and reactive–diffusive processes [15] to highlight a few.

Our main objective in this work is to present and evaluate the performance of a set of information entropy measurements, conjugated two by two, in order to characterize different classes of 3D structural patterns arising from nonlinear spatiotemporal processes. To this end, the article is organized as follows: The analytical methodology is presented in Section 2, and the data are presented in Section 3. The results, in the context of a benchmark based on the generalization of the silhouette score, are presented and interpreted in Section 4. Our concluding remarks, with emphasis on pointing out the usability of the method in the context of data-driven science, are presented in Section 6.

## 2. Methods

Various entropy metrics have been proposed in the literature, including spectral entropy, permutation entropy, and statistical complexity.

The process of defining a new metric typically involves two fundamental steps: (i) establishing the probability definition and (ii) determining the entropic form. This framework allows for the generalization of any new metric by specifying these two steps (code publicly available at https://github.com/rsautter/Eta (14 January 2024)).

In Section 2.1 and Section 2.2, we present, respectively, the key techniques for defining probabilities and entropic forms. Subsequently, in Section 2.3, we introduce a methodology to assess these metrics using criteria that are commonly applied to clustering techniques.

### 2.1. Probabilities

Probability is a concept that quantifies the likelihood of an event occurring. It is expressed as a numerical value between 0 and 1. Here, 0 signifies the complete impossibility of an event, while 1 denotes absolute certainty. Mathematically, if we consider a process with a finite number of possible outcomes, the probability Pr(E) of an event *E* is defined by the following ratio:(1)$$\mathrm{Pr}\left(E\right)=\frac{\mathrm{Number}\;\mathrm{of}\;\mathrm{favorable}\;\mathrm{outcomes}}{\mathrm{Total}\;\mathrm{number}\;\mathrm{of}\;\mathrm{possible}\;\mathrm{outcomes}}.$$

This definition is useful for gaining insight of systems that produce discrete real-valued outcomes. In such a case, a histogram of proportions of observed events is the usual tool for estimating the underlying probability distribution of such outcomes.

Many systems produce continue-valued multidimensional outcomes, and the observer needs to define methods for estimating a useful probability that is able to characterize their behavior. Approaches such as permutation and spectral analysis incorporate spatial locality and scale considerations to elucidate the occurrence of specific patterns.

In the permutation approach, local spatial differences (increase, decrease, or constancy) represent the states. New states can be generated by permuting the array elements. Thus, the probabilities account for the occurrences of those states. To extend this definition to multiple dimensions, a given array is flattened. Further details of this technique have been explored by Pessa and Ribeiro [16].

Another methodology involves spectral analysis, wherein the probability is computed as the power spectrum density (PSD) of the signal P(ω), which is normalized accordingly. Since this approach considers the probability associated with a given frequency ω, it explores the scaling relation of the signal. For instance, white noise, characterized by equal power across all frequencies, represents a type of signal exhibiting maximum entropy. In contrast, red noise presents a higher PSD for lower frequencies, leading to lower entropy values. This approach has been popularized in the literature to study time series [2,17]. The probabilities presented in this section describe the possible spatial states, while the subsequent subsection elaborates on the entropic characterization of this system.

### 2.2. Entropic Forms

Several entropy equations and generalizations have been proposed, such as Boltzmann–Gibbs entropy (also known as Shannon entropy), Tsallis entropy, and Rényi entropy. The most common form is Shannon entropy, which is expressed as follows: (2)$$S_{H}=-\sum_{i=1}^{W}p_{i}\log p_{i}.$$
Here, $p_{i}$ is the probability of state *i*, which can also comprise complex numbers [18], and *W* is the size of the set of possible events. The value of $S_{H}$ depends on the distribution. Notably, $S_{H}$ is at the maximum when all probabilities are equal, i.e., under the uniform distribution; in this case, $S_{H}=-\log W$, and it is at the minimum when $p_{i}$ is Dirac’s delta. To account for this maximum value, normalized Shannon entropy is given by the following: (3)$$S_{H}=-\frac{\sum_{i=1}^{W}p_{i}\log p_{i}}{\log W}.$$

Another significant entropic form is Tsallis entropy, proposed as a generalization of Boltzmann–Gibbs entropy [19]: (4)$$S_{q}=\frac{1-\sum_{i=1}^{W}p_{i}^{q}}{q-1},$$
where q∈R is the entropic index or nonextensivity parameter, and it plays a crucial role in determining the degree of nonextensivity in Tsallis entropy.

It is important to explore a range of values for the parameter *q* to derive a metric distinct from Shannon entropy since $\lim_{q\to 1}S_{q}=S_{H}$. Therefore, we suggest exploring values for *q* in the range of 1<q<5 and seek a relationship denoted by α, where $\log S_{q}=\alpha \log q$. This approach enables the examination of this generalization of $S_{H}$.

A unique strategy for characterizing complex nonlinear systems is gradient pattern analysis (GPA). This technique involves computing a set of metrics derived from the gradient lattice representation and the gradient moments (see Appendix A). Specifically, we highlight $G_{4}$, which is determined as the Shannon entropy from the complex representation of the gradient lattice:(5)$$G_{4}=\left|\sum_{j=0}^{V_{A}}\frac{z_{j}}{z}\ln\frac{z_{j}}{z}\right|.$$

In the lattice context, the gradient signifies the local variation of amplitudes, computed as the spatial derivative at every embedding dimension. From these spatial derivatives, the following complex representation is formed:(6)$$z_{j}=\left|v_{j}\right|e^{i\theta_{j}},$$

It comprises both the modulus ($|v_{j}|$) and phases ($\theta_{j}$). To obtain a probability, the complex notation is normalized by $z=\sum z_{j}$. For an in-depth review of this metric, please refer to [18,20]. Table 1 provides a summary of all combinations of entropic forms with associated probabilities, along with the GPA metric, that were examined in this study.

To assess the efficacy of each metric and explore the impact of various combinations of probability definitions with entropic forms, we introduce a criterion outlined in the subsequent section. This criterion is formulated with a focus on clustering the entropy measures of the dataset.

### 2.3. Silhouette Score and Generalized Silhouette Score

Non-supervised algorithms face unique challenges, and a remarkable one is defining their efficiency. The silhouette score is a criterion for defining if a set has been well clusterized [23]. Given an element $x_{i}$ in a cluster $\pi_{k}$, this metric is computed as follows [3,24]: (7)$$s\left(x_{i}\right)=\frac{b\left(x_{i}\right)-a\left(x_{i}\right)}{\max\left\{b\left(x_{i}\right),a\left(x_{i}\right)\right\}},$$
where $a(x_{i})$ is the average dissimilarity, which is the average distance of $x_{i}$ to all other elements in the cluster $\pi_{k}$, and $b(x_{i})$ is the average distance to the elements of other clusters. The greater the $s(x_{i})$ value, the better performance of the clustering algorithm because it has produced groups with low dissimilarities and large distances between clusters. This technique can be extended to feature extractions if one considers the individual datasets as the clusters $\pi_{k}$. However, it is equally essential to account for the potential correlation between metrics, as metrics may inadvertently capture the same data aspects, which is undesirable. To mitigate this, we use the modulus of the Pearson correlation |r| to form the penalty term 1−|r| as follows: (8)$$s^{\prime }\left(x_{i}\right)=\left(1-|r|\right)\frac{b\left(x_{i}\right)-a\left(x_{i}\right)}{\max\left\{b\left(x_{i}\right),a\left(x_{i}\right)\right\}},$$
which we call the generalized silhouette score (GSS).

After defining a group of entropy measurements and the tool (GSS), which allows the determination of the best pair of measurements to compose a 2D parameter space, we selected the dataset to test and validate our methodological approach.

## 3. Data

Our main objective is to test the performance of a space composed of two entropy measures in which it is possible to distinguish different classes of complex spatiotemporal processes. For this first study, we chose turbulence-related processes and simulated dynamic colored noises.

We employ simulated data related to the following processes: (i) white noise; (ii) colored noise; (iii) weak turbulence; (iv) hydrodynamic turbulence; and (v) magnetohydrodynamic turbulence (MHD). The main reason for choosing these processes, except colored noise, is that they all present random-type patterns with underlying dynamic characteristics based on physical processes described by partial differential equations (diffusion, reaction, and advection). Each was obtained from simulations identified in Table 2.

Based on the power-law-scaling algorithm technique [25], we created our noise simulator [26]. The data representing weak turbulence (also called chemical or reactive–diffusive turbulence) were obtained from the solution of the Ginzburg–Landau complex equation [15,27]. The hydrodynamic turbulence patterns were selected from the John Hopkins database (JHTDB) [28], and the MHD turbulence was simulated using the PENCIL code [12]. Details regarding the simulations are provided in the Supplementary Materials in the GitHub repository.

To test the approach based on entropy measurements, we selected a total of 25 snapshots representing the evolution of each chosen process. After selecting the middle slice of the hypercube, we uniformly resized all snapshots to 64×64 byte-valued pixels using nearest neighbor interpolation; while this resizing expedites the analysis, it does entail a loss in resolution. The snapshots were extracted from 3D simulations, taking the analysis of the central slice of each hypercubeas a criterion as the measurement technique used to act on matrices within a two-dimensional approach.

Figure 1 shows representative snapshots of the respective spatiotemporal processes. These visualizations provide a compelling narrative of the dynamic behavior of each system, highlighting the wide variety of patterns that emerge through temporal dynamics in the phase space.

The numerical procedures and/or technical acquisition details related to the data shown in Figure 1 are available in the Supplementary Materials in the repository (https://github.com/rsautter/Eta/ (14 January 2024)) and in the section entitled “Data Simulations”.

## 4. Results and Interpretation

The analyses in this study were conducted within 2D metric spaces, encompassing all possible entropy measure combinations. Based on the minimum information principle, this configuration offers advantages in terms of interpretability, considering the minimum set of parameters that can be addressed as labels within a possible machine learning interpretation. Our approach to measuring entropies from the data follows the following steps:Input of a snapshot;Pre-processing for which its output is a 64×64 matrix with amplitudes ranging from 0 to 255;Generation of three matrix data outputs: 2D histogram, 2D permutation, and 2D FFT spectra;For each of the three domains, the entropy measures are calculated.

Given the definition of the three types of domains interpreted as probabilities (from histogram, permutation, and spectrum), we have six entropy variations, as detailed in Section 2. To distinguish these metrics, we introduced superscripts denoted by *h* for histogram probability, *p* for permutation probability, and *s* for spectral probability. The GPA analysis yields another metric, resulting in 21 scores, as illustrated in Figure 2.

As a result, the most effective combination is the following pair: spectral Tsallis entropy ($S_{q}^{s}$) and Shannon permutation entropy ($S_{H}^{p}$). A visual representation of this space, accompanied by some snapshots, is presented in Figure 3. In this space, the metrics reveal a constant Shannon permutation entropy dynamical noise system, which is solely distinguished by spectral Tsalllis entropy, indicating the differences in the scaling effects in pattern formation. Conversely, the distinct complex nonlinear characteristics and reaction terms observed in MHD simulations are more pronounced in Shannon permutation entropy, accentuating the diversity of localized patterns alongside the larger-scale ones.

The analysis of entropy distribution is essential in a classification context, as it offers insights into the similarity between a new dataset and various models. However, carefully analysing the entropy metrics over time can highlight important aspects of the underlying physical processes. For instance, the transition from initial conditions to an oscillatory relaxation state is evident in Figure 4. This outcome aligns with expectations in the context of the CGL system due to the periodic nature of the reaction term. However, it is essential to highlight that in this introductory study, we avoided simulations with more complex regimes (such as relaxations) as the primary purpose here is to present a new method, and the objective here is not to use it to deepen the physical interpretation of each process.

## 5. Outlook

Based on the study and approach presented here, we defined a methodological pipeline for the spatiotemporal characterization of simulated and/or observed complex processes (Figure 5). The method can be applied to identify and segregate different classes of processes and to classify isolated patterns when necessary. In a context where measured and simulated data may exist, it also serves to validate models. Likewise, the pair of entropy measurements can also serve as a binomial label for training deep learning architectures for automatic classification.

## 6. Concluding Remarks

This work carried out a comprehensive analysis of entropy metrics and their application to complex extended nonlinear systems. The study explored new approaches, including different entropy measures and a new *generalized silhouette score* for measurement evaluation.

Through the meticulous consideration of *canonical* datasets, distinct patterns have been characterized in terms of entropy metrics. The pivotal finding was the identification of the optimal pair: spectral Tsallis entropy ($S_{q}^{s}$) and Shannon permutation entropy ($S_{H}^{p}$), yielding superior outcomes in the generalized silhouette score. This combination showcased efficacy in distinguishing spatiotemporal dynamics coming from different classes of turbulent-like processes, including pure stochastic 2D $1/f^{-\beta }$ (colored) noise.

The new method contributes valuable insights into applying entropy probabilistic measures, providing a foundation for future studies in terms of extended complex system pattern formation characterization.

Initial work considering entropy measurements for training machine learning models is underway. In this context, it also includes a study of the computational complexity of the method for a benchmark with other measures and approaches that may emerge. This strategy is fundamental when we think about the presented method being applied in a data science context.

## Figures and Tables

**Figure 1 entropy-26-00508-f001:**
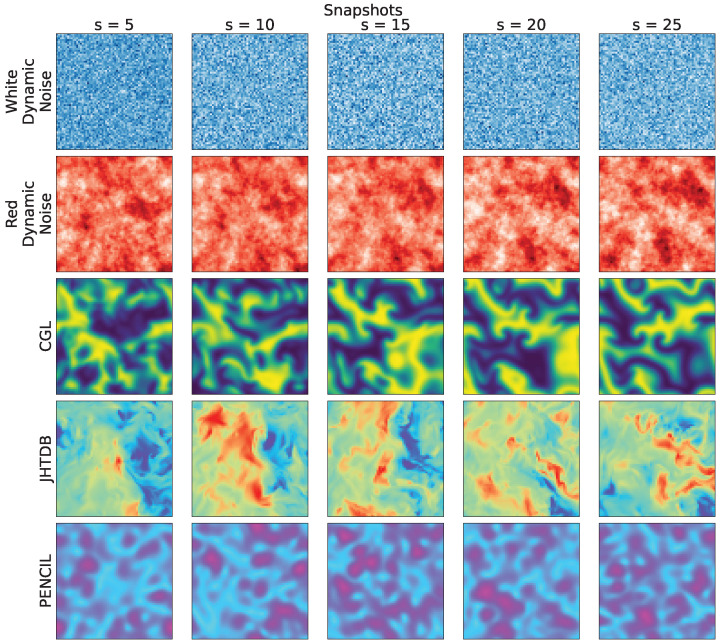
Snapshots of the spatiotemporal evolution of each selected system class, listed in Table 2. Each row shows one of the simulations, rendered at time steps that show representative pattern dynamics: dynamic white noise (β=0 represented by colormap ‘Blues’) on the 1st row; random red noise (β=2, represented by colormap ‘Reds’) on the 2nd row; weak turbulence from the reaction–diffusion complex Ginzburg–Landau dynamics on the 3rd row (represented by colormap ‘viridis’); fully developed turbulence from JHTDB on the 4th row (represented on colormap ‘rainbow’) and MHD turbulence from PENCIL on the 5th row (represented by colormap ‘cool’).

**Figure 2 entropy-26-00508-f002:**
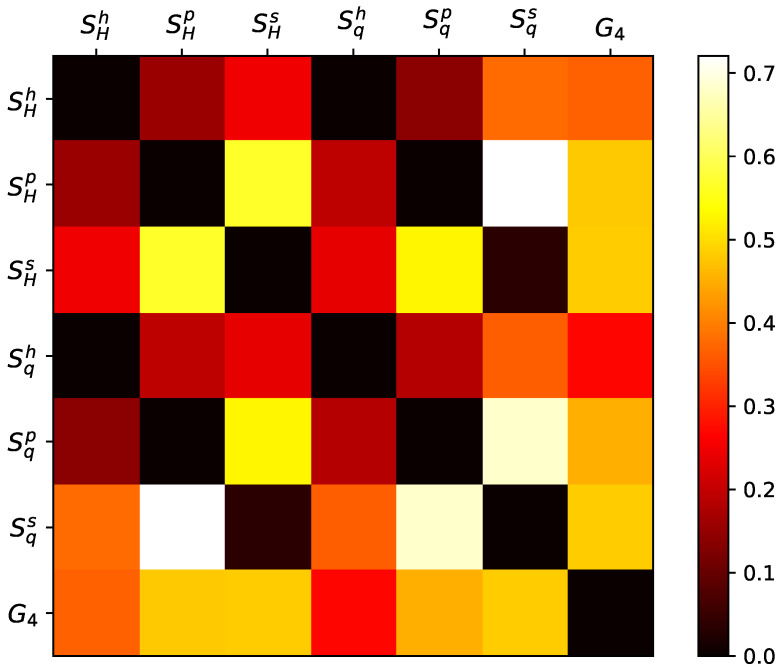
Generalized silhouette score for all 2D metric combinations. Higher values on the heatmap indicate superior metric performance. The optimal result is achieved with the pairing of spectral Tsallis entropy and Shannon permutation entropy ($S_{q}^{s}\times S_{H}^{p}$).

**Figure 3 entropy-26-00508-f003:**
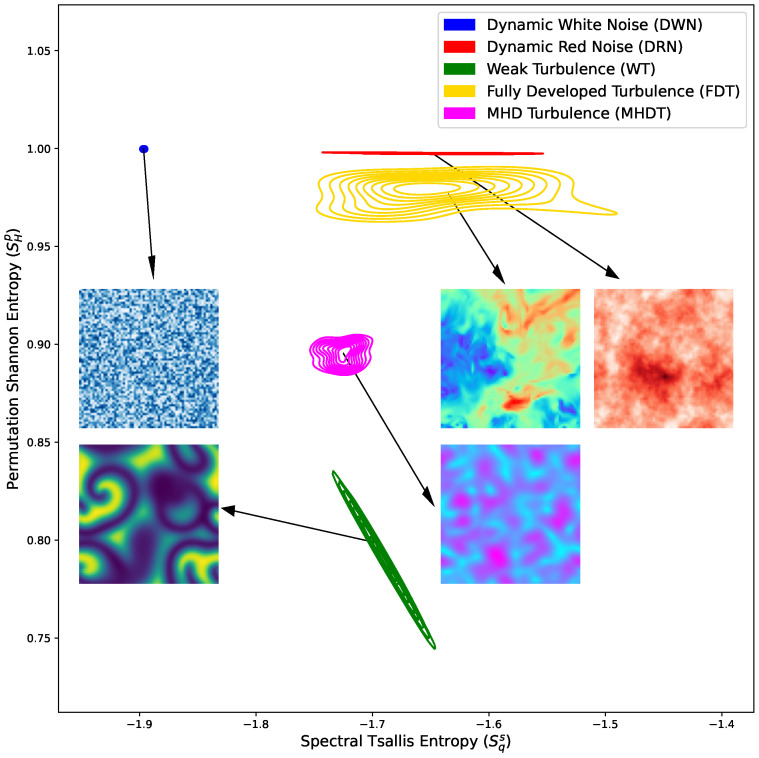
Optimal outcomes achieved are assessed through the generalized silhouette score criterion. The method achieves its best performance in the ($S_{q}^{s}\times S_{H}^{p}$) parameter space.

**Figure 4 entropy-26-00508-f004:**
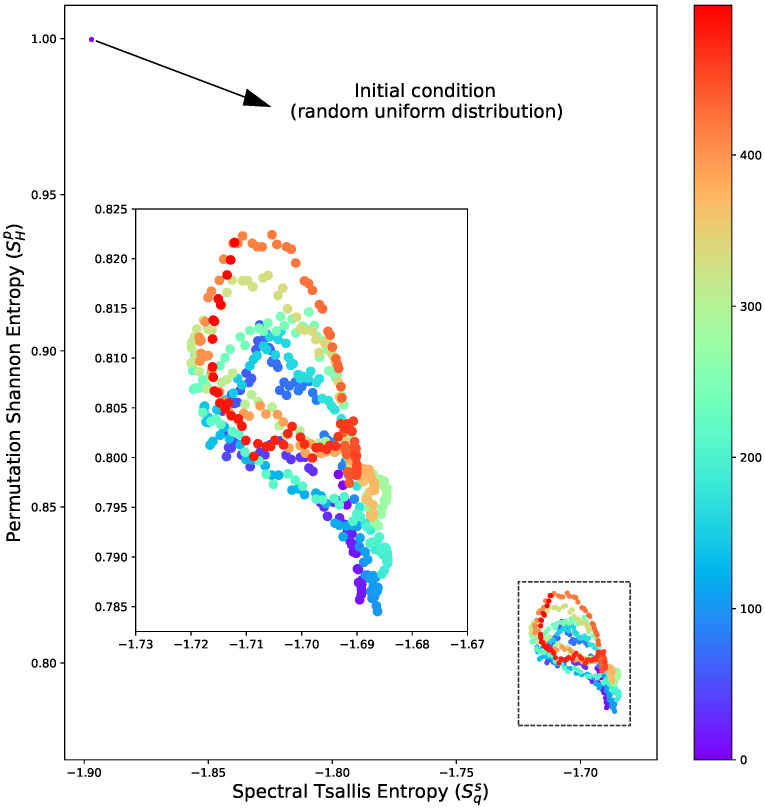
Best entropy set according to the generalized silhouette score (see Figure 2) for the 3D-CGL solution over time, where the oscillatory dynamic of the system is highlighted. The color indicates the snapshot, where 500 samples are presented.

**Figure 5 entropy-26-00508-f005:**
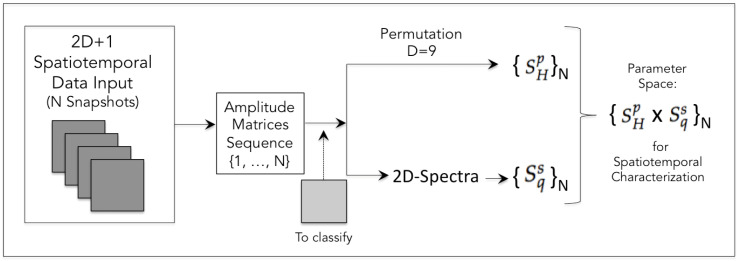
Pipeline of the method proposed in this study based on the best results found: A sequence of snapshots from the simulation of a given process (in the 2D+1 or 3D+1 domains) comprises the input from which entropy measurements will be obtained. To calculate the respective Shannon permutation entropy values $S_{H}^{p}$, the permutation values are obtained (see Appendix B). To calculate the spectral Tsallis entropy $S_{q}^{s}$, the respective spectra are obtained. From the calculated values, the parameter space is constructed where where it is proposed to characterize the underlying process. The space also works for classifying isolated patterns, taking as reference the distinct processes that have already been characterized.

**Table 1 entropy-26-00508-t001:** Entropy measures.

Measure	Probability	Entropic Form	Reference
$S_{H}^{h}$	histogram	Shannon, Equation (Equation 3)	Lesne [21]
$S_{H}^{p}$	permutation	Shannon, Equation (Equation 3)	Pessa, Ribeiro [16]
$S_{H}^{s}$	spectral	Shannon, Equation (Equation 3)	Abdelsamie et al. [9], Abdullah et al. [3]
$S_{q}^{h}$	histogram	Tsallis *q*-law, Equation (Equation 4)	Li and Shang [22]
$S_{q}^{p}$	permutation	Tsallis *q*-law, Equation (Equation 4)	Li and Shang [22]
$S_{q}^{s}$	spectral	Tsallis *q*-law, Equation (Equation 4)	This paper
$G_{4}$	gradient	Complex Shannon, Equation (Equation 5)	Ramos et al. [18]

**Table 2 entropy-26-00508-t002:** Datasets and references.

Simulation	Process	Reference
White Dynamic Noise	Spatiotemporal stochastic	Timmer et al. [25]
Red Dynamic Noise	Spatiotemporal stochastic	Timmer et al. [25]
CGL ^1^	Weak turbulence	Sautter [26], Sautter et al. [27]
JHTDB	Fully developed turbulence	Brandenburg et al. [12]
PENCIL	MHD turbulence	Brandenburg et al. [12]

^1^ Our 3D simulator is public available at https://github.com/rsautter/Noisy-Complex-Ginzburg-Landau (14 January 20224).

## Data Availability

All the mathematical content and data used in this work in a GitHub repository (https://github.com/rsautter/Eta/ (14 January 2024)) to guarantee the reproducibility of this experiment.

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
