# Peer review of "Characterizing Complex Spatiotemporal Patterns from Entropy Measures"

_entropy, 2024, doi:10.3390/e26060508_

Round 1
Reviewer 1 Report
Comments and Suggestions for Authors
I have read the very interesting submitted manuscript. I have some questions of fundamental nature that need to be discussed. A discussion on the "similarity" of the data is needed. I would claim that there are fundamental differences in the nature of the data. Often fluid turbulence/MHD turbulence modelled by stochastic methods but are they necessarily stochastic? The fundamental equations are deterministics however they lead to chaotic behaviour that seems stochastic due to our lack of more fundamental understanding. Thus I would claim that there are differences in the used data. In comparison dynamic colored noise is stochastic and of a fundamentally different nature. Please discuss the implications for this work?
Another point that should be discussed is the claim that the range, 1<q<5 should be explored. This could be connected much deeper to the meaning of the nonextensivity parameter q, see Anderson Phys Plasmas 2014, 21, 122101.
The description of the data should be improved. Please provide input data used to generate the simulations. Are they 2D or all 3D? How are the snapshots generated?
It is widely different physics present in 3D fully developed turbulence vs 2D, similar in the case of MHD. Also, in this regard the fully stochastic data should be significantly different.
Comments on the Quality of English Language
Only minor check of language is needed.
Author Response
R1_1. I have read the very interesting submitted manuscript. I have some questions of fundamental nature that need to be discussed. A discussion on the "similarity" of the data is needed. I would claim that there are fundamental differences in the nature of the data. Often fluid turbulence/MHD turbulence is modelled by stochastic methods but are they necessarily stochastic? The fundamental equations are deterministic however they lead to chaotic behavior that seems stochastic due to our lack of more fundamental understanding. Thus I would claim that there are differences in the used data. In comparison, dynamic colored noise is stochastic and of a fundamentally different nature. Please discuss the implications of this work.
We thank the reviewer for raising this point. We clarify that the selected spatiotemporal patterns, coming from physical models, to test the proposed methodology, are all related to turbulent-like processes. In particular, the CGLE is a canonical model for modeling the so-called "weak turbulence". The stochasticity inherent to them comes from the unpredictability imposed by the non-linearity and diffusive component common to all. In this application, therefore, the main objective is to bring a new tool that is capable of classifying different turbulent dynamics, from a spatiotemporal pattern analysis domain, without extrapolating to a particular interpretation of the underlying physical processes. It is important to highlight that Noise Patterns were added, not as physical models, but because they are generally used as an initial condition in most simulations, which also impose a certain degree of stochasticity on the 3D+1 dynamics throughout its evolution.
R1_2. Another point that should be discussed is the claim that the range, 1<q<5 should be explored. This could be connected much deeper to the meaning of the nonextensivity parameter q, see Anderson Phys Plasmas 2014, 21, 122101.
We thank the reviewer for this issue as well. It is important to clarify that the parameter "q", and the corresponding entropy, in our approach are used as a purely mathematical entity. To achieve this, we created the criterion of choosing a significant range of q to extract the entropy measure that acts as a classifier. The thermodynamic concepts of "non-extensivity" and "non-additivity" associated with q are not relevant to this application.
R1_3 The description of the data should be improved. Please provide input data used to generate the simulations. Are they 2D or all 3D? How are the snapshots generated?
The data utilized in this study was obtained through established techniques validated within the scientific community, as outlined in Table 2. All simulations conducted are in 3D+1 format, indicating that for each timestamp, a three-dimensional cube of data is generated. To assess the performance over time, we opted to analyze the middle slice of the data cube, as described in detail in line 168 of the manuscript. The Python code used for the entire experiment, including examples and datasets, is available in the repository linked after Section 3. This repository contains the data utilized in our study. Additionally, adjustments were made to line 47 for clarity.
R1_4 It is widely different physics present in 3D fully developed turbulence vs 2D, similar in the case of MHD. Also, in this regard, the fully stochastic data should be significantly different.
We agree with this reviewer's statement. In fact, this issue is even more fundamental than comparing MHD with neutral fluid turbulence in different dimensions. In this application, for reasons of computational complexity and following the principle of minimum information, we decided to analyze 2D slices from 3D simulations. In a future application, it would be interesting to generalize the application of the method directly to hypercubes (3D) and compare the classification results of the generated patterns. The same reasoning applies to generating only 2D simulations and repeating the analysis on them. But this approach will be left for future work. We are including this discussion in the concluding remarks.

Reviewer 2 Report
Comments and Suggestions for Authors
Comments on the paper entitled:
"Characterizing Spatiotemporal Complex Patterns from Entropy Measures"
The text is well-written and acknowledges the need for further development of entropy measurements in understanding complex systems, particularly in 2D+1 and 3D+1 dimensions. However, in my opinion, some parts of the paper suffer from technical weaknesses and could be improved for better flow and clarity before being accepted for publication. Moreover, some technical issues should be reviewed and addressed. Here are the detailed remarks:
Text technical issues:
1. The notation "2D + 1" and "3D + 1" likely refer to spatial dimensions plus time. D also refers to the embedding dimension in Permutation Entropy and is fixed at 9.
2. The abstract mentions four classes of similar processes related to the evolution of random patterns. However, later in the text, it refers to "the five selected classes," which might suggest the inclusion of an additional class not explicitly listed. This discrepancy could be a mistake in the writing. If the intention is to refer to four classes, then it should be consistent throughout the text.
3. The parameter (beta) used to simulate red noise should be indicated in Fig.1. I couldn’t find parameters for simulations in the repository that the authors indicate.
4. Why is D=9 relevant for the presented analysis? Did the authors verify other values of D?
5. What is the size in pixels of the snapshots? The size is crucial for selecting the parameter of the embedding dimension D. D=9 appears to be too large since it requires estimating an ordinal probability over factorial 9 possibilities, which may result in biased and poor estimation of permutation entropy (Figure 3).
6. What kind of pre-processing do you apply to transform the snapshot to 64x64 pixels? What is the impact of such pre-processing on the metrics you are using?
7. Mathematical notations in Appendices need to be verified: Check G_4 in eq5 and eqA4. Check the symbol sum for i=0 to V_A in both eqA2 and eqA3. Check the sum symbol in eqA5. What is v_i^A in line 255? Define \phi_i in line 255. In line 255, “i” is an index, but “i” also denotes the complex number i^2=-1.
8. Define r in eq8.
9. How do you calculate spectra? Which parameters do you use to obtain the spectra estimation? The same question applies to histograms.
10. Only two entropy measures were tested: Tsallis and Permutation entropy. They are applied to different probability distributions (histogram, spectra, and permutation). I assume that permutation means ordinal patterns. This contradicts the seven possible ways announced in the abstract to measure entropy from a matrix.
11. Main Technical issues: The abstract announcement states, “Notably, our observations reveal the segregation of reaction terms in this [..] space, a result that identifies specific sectors for each class of dynamic process and can be used to train machine learning models for automatic classification of complex spatiotemporal patterns”. However, in lines 207-211: “However, it is essential to highlight that in this introductory study, we avoided simulations with more complex regimes (such as relaxations) as the primary purpose here is to present a new method, and the objective here is not to use it to deepen the physical interpretation of each process. There is no new method, only two classical existing entropy measures applied to datasets in time and frequency domains. The GSS defined using the Pearson correlation is not motivated. Other GSS were introduced in literature, do the authors try them? Moreover, as I said before, authors must justify the use of D=9 for PE.

Author Response
R2_1. The notation "2D + 1" and "3D + 1" likely refer to spatial dimensions plus time. D also refers to the embedding dimension in Permutation Entropy and is fixed at 9.
We express our gratitude to the referee for their appointment. We change the notation of the embedding dimension to $d$ as it has been used in other publications, e.g.: https://www.nature.com/articles/s42005-021-00696-z.
R2_2. The abstract mentions four classes of similar processes related to the evolution of random patterns. However, later in the text, it refers to "the five selected classes," which might suggest the inclusion of an additional class not explicitly listed. This discrepancy could be a mistake in the writing. If the intention is to refer to four classes, then it should be consistent throughout the text.
We agree with the reviewer's suggestion and have accordingly revised the categorization to encompass five data types. In our revised schema, we classify the two forms of noise—white and red—as a singular category, albeit with two distinct subtypes. Nonetheless, given that each subtype undergoes individual analysis, we acknowledge the merit of designating them as separate classes within the data taxonomy.
R2_3. The parameter (beta) used to simulate red noise should be indicated in Fig.1. I couldn’t find the simulation parameters in the repository that the authors indicate.
Beta terms were introduced in the figure as suggested.
R2_4. Why is D=9 relevant for the presented analysis? Did the authors verify other values of D?
We used the minimum size of a squared window with a central pixel. Larger windows would increase the computational cost. The chosen D=9 was enough to provide a good description of the processes.
R2_5. What is the size in pixels of the snapshots? The size is crucial for selecting the parameter of the embedding dimension D. D=9 appears to be too large since it requires estimating an ordinal probability over factorial 9 possibilities, which may result in biased and poor estimation of permutation entropy (Figure 3).
We thank the reviewer for raising this point and here we answer 4 and 5. We employ the $d_x \times d_y = 9$ as it represents the minimal kernel size encompassing a central pixel, a crucial aspect from the perspective of central analysis. Acknowledging the reviewer's observation regarding the lack of clarity on this matter within the text, we have taken the initiative to incorporate this detail into the abstract section for enhanced comprehensibility.
R2_6. What kind of pre-processing do you apply to transform the snapshot to 64x64 pixels? What is the impact of such pre-processing on the metrics you are using?
Due to variations in simulation sizes based on different applications, we standardized the size of the snapshots for our study. “After selecting the middle slice of the hypercube, we uniformly resized all snapshots to 64x64 dimensions using nearest neighbor interpolation while this resizing expedites the analysis, it does entail a loss in resolution”. We agree with the revisor that was not explicit in the methodology and we included that in the manuscript in line 165
R2_7. Mathematical notations in Appendices need to be verified:
- Check G_4 in eq5 and eqA4. Alterei a equação do Anexo e coloquei no mesmo do texto correndo em j
- Check the symbol sum for i=0 to V_A in both eqA2 and eqA3. Feito
- Check the sum symbol in eqA5. Feito
- What is v_i^A in line 255? Define \phi_i in line 255. In line 255, “i” is an index, but “i” also denotes the complex number i^2=-1.Feito
We thank you for the thorough analysis. All changes were made as suggested.
R2_8. Define r in eq8.
The sentence where we wrote eq8 defines r as the Person Correlation in line 148, which is a common analysis used in statistics
R2_9. How do you calculate spectra? Which parameters do you use to obtain the spectra estimation? The same question applies to histograms.
In data analysis, it's a common practice to employ windows to enhance the visualization of power spectra. However, when it comes to calculating entropy, spectra are typically derived from the power of Fourier Transform indices. Utilizing windows in Fourier transform necessitates a meticulous examination of each case, which complicates the establishment of parameters for spectrum calculation. Conversely, the computation of histograms relies on defining the number of bins. Adopting a robust approach, histograms are generated using the standard method provided by the numpy library.
R2_10. Only two entropy measures were tested: Tsallis and Permutation entropy. They are applied to different probability distributions (histogram, spectra, and permutation). I assume that permutation means ordinal patterns. This contradicts the seven possible ways announced in the abstract to measure entropy from a matrix.
The term permutations and ordinal patterns are interchangeable in the literature see https://www.nature.com/articles/s42005-021-00696-z. Indeed, two types of entropy combined with three probability distributions result in six distinct analytical methods. Furthermore, we are incorporating the use of $G_4$, thereby introducing a seventh method to our analysis.
R2_11. Main Technical issues: The abstract announcement states, “Notably, our observations reveal the segregation of reaction terms in this [..] space, a result that identifies specific sectors for each class of dynamic process and can be used to train machine learning models for automatic classification of complex spatiotemporal patterns”. However, in lines 207-211: “However, it is essential to highlight that in this introductory study, we avoided simulations with more complex regimes (such as relaxations) as the primary purpose here is to present a new method, and the objective here is not to use it to deepen the physical interpretation of each process. There is no new method, only two classical existing entropy measures applied to datasets in time and frequency domains. The GSS defined using the Pearson correlation is not motivated. Other GSS were introduced in literature, do the authors try them? Moreover, as I said before, authors must justify the use of D=9 for PE.
In fact, the Shannon equation and Tsallis equation are well-established in the literature. However, the combination of the spectral probability with the scaling law of Tsallis entropy hasn’t been presented in the literature as we explicitly show in Table 1. While Puri et al. (2021) introduced an analysis employing a spectral Tsallis approach at ICCICT, the specific values or range for the q parameter of Tsallis entropy were not delineated by the authors.
The clustering process and the evaluation of feature extraction techniques are similar, but they have nuances that need to be considered. The clustering measure only considers the ability to segregate points in a space, while in the feature extraction process, the resulting measures should be independent. Due to this requirement, we include the statistical correlation measure as a penalty in the metric.

Round 2
Reviewer 1 Report
Comments and Suggestions for Authors
I have read the paper and response and can now recommend it for publication.